# Immunohistochemical Expression of CD44, MMP-2, MMP-9, and Ki-67 as the Prognostic Markers in Non-Clear Cell Renal Cell Carcinomas—A Prospective Cohort Study

**DOI:** 10.3390/jcm11175196

**Published:** 2022-09-02

**Authors:** Magdalena Chrabańska, Magdalena Rynkiewicz, Paweł Kiczmer, Bogna Drozdzowska

**Affiliations:** Department and Chair of Pathomorphology, Faculty of Medical Sciences in Zabrze, Medical University of Silesia, ul. 3 Maja 13-15, 41-800 Zabrze, Poland

**Keywords:** CD-44, MMP-2, MMP-9, Ki-67, renal cell carcinoma, papillary renal cell carcinoma, chromophobe renal cell carcinoma

## Abstract

CD44 is the most frequently reported marker of the cancer stem cells in renal cell carcinoma (RCC). Matrix metalloproteinases MMP-2 and MMP-9 are key regulators of tumor invasion and metastasis. The aim of this study was to investigate the clinicopathologic and prognostic values of the immunohistochemical expression of CD44, MMP2, MMP9, and Ki-67 in papillary and chromophobe RCCs. In the case of papillary RCC, MMP-2 expression was positively correlated with patient age (*p* < 0.05), while CD44 expression was positively correlated with tumor stage (τ = 0.26, *p* < 0.05). Moreover, CD44 expression positively correlated with MMP-9 (τ = 0.39, *p* < 0.05). In the case of chromophobe RCC, only Ki-67 expression was negatively correlated with tumor stage (τ = −0.44, *p* < 0.05). During follow-up, a death was documented in 6 patients with papillary RCC. In these patients, CD44 expression was not a significant factor affecting the overall survival of patients (*p* > 0.05), whereas there was a positive correlation between increased MMP-9 expression and shorter overall survival (*p* < 0.05). Taken together, carcinogenesis in papillary RCC is probably dependent on both cancer stem cells and metalloproteinases activity. Expression of CD44 and MMP-9 can significantly improve the prediction of papillary RCC prognosis in the future.

## 1. Introduction

In the following years, the clinical spectrum of renal epithelial tumors has expanded by the increased recognition of new tumors named by their architectural pattern, anatomic location, or associated diseases. The fourth edition of the World Health Organization (WHO) classification of urogenital neoplasms account for more than 50 entities and provisional entities of which clear cell (ccRCC), papillary (pRCC), and chromophobe (chRCC) tumors account for 65–70%, 18.5%, and 5–7%, respectively. Other RCC subtypes are rare [1,2,3]. Every subtype is associated with different biologic behavior, prognosis, treatment options, and response to therapy, therefore knowledge of each RCC subtype is important [2,4]. Although, in most cases, the histopathological diagnosis is based on routine morphological examination, immunohistochemical analysis is a very useful ancillary technique in several contexts, including subtyping of RCCs, diagnosing rare types and unusual cases of renal neoplasms, and predicting the biologic behaviour and prognosis of the renal tumors. To date, the traditional assessment of prognosis for RCC patients relies primarily on histological features, nuclear grade, and tumor stage [5]. However, these parameters are neither reliable nor precise for predicting the biologic behaviour [6,7]. The vast majority of studies have explored the biologic behaviour and prognostic markers in the context of ccRCC. For the patients with non-clear cell histology, there is little consensus because there is not enough available data among the various subtypes of RCC. Herein, we have investigated the potential parameters which can be virtually helpful for predicting the biologic behaviour of papillary and chromophobe RCCs.

Ki-67 is a non-histone protein that is absent in non-dividing cells, while it is present during the entire cell cycle [8]. This feature makes it a great marker for determining the proliferation fraction of tumor cells. Proliferation index of RCC as determined by Ki-67 is proved to be the measure of aggressiveness of a tumour and it is strongly correlated with nuclear grade and clinical survival in RCC.

Tumor metastasis involves strong interactions between the invading cancer cells and tumor microenvironment cells, which produce specialized proteolytic enzymes promoting degradation of the extracellular matrix (ECM). Among these enzymes, urokinase and a variety of matrix metalloproteinases (MMPs), a family of zinc and calcium-dependent proteolytic enzymes, degrade different components of ECM including collagen, laminin, fibronectin, vitronectin, elastin, and proteoglycans. The MMP group includes more than 20 different enzymes. On the basis of their structure, cell localization, and substrate specificity, MMPs are divided into several groups such as collagenases, gelatinases, stromelysins, matrilysins, membrane-type MMPs, and other MMPs according to their catalytic activity. MMPs take part in invasion, migration, metastasis, and tumorigenesis [9,10]. Among the MMPs, type IV collagenases MMP-2 (72-kDa gelatinase A) and MMP-9 (92-kDa gelatinase B) are associated with cell growth, migration, invasion, inflammation, and angiogenesis by digesting targets such as type IV collagen—a basic structural component within the ECM and basement membrane [9,11,12,13]. A significant association between tumor aggressiveness and increased MMP-2 and MMP-9 levels has been proved [12,13,14,15].

Cancer stem cells are capable of self-renewal and are responsible for cancer progression, recurrence, distant metastasis, and chemo- and radioresistance in many malignancies [16]. CD44, a cell surface integral membrane glycoprotein, is expressed in cancer cells and is admitted as a molecular marker for cancer stem cells [8]. As a protein intervened in cell motility, CD44 has been utilized as a marker for tumor aggressiveness [17,18,19]. Nevertheless, there are contradictory reports in the case of RCC.

The aim of our study was to investigate the correlations between the proliferation index determined by Ki-67, expression of CD44, MMP2, and MMP9 and analyze the interactions between the cancer stem cells and metalloproteinases as potential prognostic markers in pRCC and chRCC. We also compared the clinicopathological and prognostic values with CD44, MMP2, MMP9, and Ki-67 expression to provide a better insight of the course of rare types of RCC.

## 2. Materials and Methods

### 2.1. Patient Characteristics and Tumor Samples

A total of 59 paraffin-embedded tissues from non-ccRCC samples were included in this study. The study group consisted of 41 cases of pRCCs and 18 cases of chRCCs searched from June 2015 to October 2020.

All specimens had been obtained during partial or radical nephrectomy. In all cases, the submitted surgical specimens were handled according to the current guidelines of the Polish Society of Pathologists and complied with the recommendations of the ISUP and the WHO for specimen handling, sampling, and reporting [20,21]. The tissue specimens were formalin-fixed and paraffin-embedded using a routine pathological tissue processing technique. The hematoxylin- and eosin-stained slides from all cases were reviewed by two pathologists who assigned both a WHO/ISUP grade and eighth edition of the American Joint Committee on Cancer (AJCC) TNM pathological staging category [22]. All sections were then assessed for: morphotype, tumor size, WHO/ISUP grade (rated only for pRCC), the presence of necrosis, sarcomatoid and rhabdoid differentiation, small vessel lymphovascular invasion, neuroinvasion, fibrous renal capsule invasion, perinephric fat invasion, renal sinus fat and vascular invasion of renal sinus vessels, macroscopic main renal vein invasion, and AJCC TNM pathologic stage of the primary tumor (pT). Follow-up data included: date of nephrectomy, survival status, date of death, and/or date of last follow-up.

This study was approved by the Institutional Review Board of Medical University of Silesia, Katowice, Poland (KNW/0022/KB/228/19). Patient data were kept fully anonymous in all steps.

### 2.2. Immunohistochemical Staining

For each case of pRCC and chRCC, a representative slide of the tumor and the corresponding paraffin block were selected. Four-micron sections were cut on glass slides and air-dried during the night. Following deparaffinization in xylene and rehydration in alcohol, heat-induced epitope retrieval was achieved by immersing slides in buffer EnVision FLEX Target Retrieval Solution (Perlan, Beaverton, OR, USA) (pH 6.0 or 9.0) and boiling at 95 °C for 20 min. Then, slides were pre-incubated with blocking solution EnVision FLEX PEROXIDASE-BLOCKING REAGENT (Perlan, Beaverton, OR, USA) for 5 min. The staining was performed in an automated immunostainer according to the manufacturer’s instructions (Table 1). After this step, the sections were rinsed in buffer EnVision FLEX Wash Buffer (Perlan, Beaverton, OR, USA) and incubated with visualization system EnVision FLEX/HRP (Perlan, Beaverton, OR, USA) for 20 min. Staining patterns were visualized by exposure to 3, 3′-diaminobenzidine DAB (Perlan, Beaverton, OR, USA) to achieve visualization of the antigens and counterstaining with hematoxylin. Finally, the slides were dehydrated in alcohol, cleared in xylenes, and mounted for examination. In each run of the experiment, replacement of the primary antibody with Tris-buffered saline was used as a negative control. The human tonsillar tissue was used as a positive control for Ki-67, CD44, and MMP-9, while tissue of invasive urothelial carcinoma was a positive control for MMP-2.

### 2.3. Evalualtion of Immunostaining

Semi-quantitative analysis was performed to evaluate the CD44, MMP-2, and MMP-9 expression. Staining pattern was heterogeneous and ranged from staining of a small number of tumor cells with low intensity to strong and diffuse positivity throughout the tumor (Figure 1 and Figure 2). CD44 immunoreactivity was noted as membranous type, while staining with MMP-2 and MMP-9 was present in the cytoplasm in the form of granules of various sizes and staining intensity. The modified Allred et al. method was used to evaluate both the intensity and the proportion of immunohistochemical staining [9,23]. The intensity scores ranged from negative to strong as follows: 0 = negative, 1 = weak, 2 = moderate, and 3 = strong. The proportion scores ranged from 0 to 5 and was categorized according to the positive tumor cells as follows: 0 = no staining, 1 = up to 1/100 positive cells, 2 = 1/100 to 1/10 positive cells, 3 = 1/10 to 1/3 positive cells, 4 = 1/3 to 2/3 positive cells, 5 = >2/3 positive cells. To calculate the total immunohistochemical score, the proportion and intensity scores were multiplied for each specimen (range from 0 to 15) [24]. Then, overall immunohistochemical scores were classified into three groups as follows: 0–5 as Group 1 (low expression), 6–10 as Group 2 (moderate expression), and 11–15 as Group 3 (high expression).

For quantitative assessment of Ki-67 expression, first hot spots were determined using low power and then approximately 1000 cells were counted in 5 high power fields. Percentage of positive nuclei was determined by dividing the number of cells stained by the total number of cells per high-power field (HPF) and multiplying by 100.

The immunohistochemical staining results were examined by two independent pathologists, who were blinded to the patients’ clinicopathological details. Inconsistency in the scores were discussed to reach a consensus for all samples and in order to avoid bias.

### 2.4. Statistical Analysis

All analyses were performed using STATISTICA software (Statsoft, Tulsa, AK, USA) and R Language in Rstudio Environment. Quantitative data was presented as mean ± SD and compared using *t*-test if the normality assumptions was fulfilled. Normality of the data was confirmed by a Shapiro–Wilk test. In the case of non-normal distribution, non-parametric Mann–Whitney U test was performed. Pearson’s chi-squared (Chi^2^) test was used to determine differences between qualitative variables. Correlation analysis was performed using Kendall’s Tau correlation coefficient. Survival analysis was performed using Kaplan–Meier estimator with Gehan-Wilcoxon test with Mantel correction for multiple groups to determine differences between groups. Effect size was calculated for each test, using Cramer’s V for Chi^2^ tests and Cohen’s d for Mann–Whitney U test and *t*-test. Basing on obtained effect size, power of each test was calculated using G*Power software or pwr package for R language in Rstudio software. *p*-values of <0.05 were considered to be statistically significant.

In reporting this study, the standard methodology was reported according to “Strengthening the Reporting of Observational Studies in Epidemiology” (STROBE) guidelines [25].

## 3. Results

### 3.1. Clinicopathological Data

The sample population included a total of 59 non-clear cell renal cell carcinoma (non-ccRCC) patients. The RCC subtypes, pRCC (*n* = 41, 69.50%) and chRCC (*n* = 18, 30.50%) were represented in the sample population. The mean age of the patients was 63.19 (SD = 9.74) years, ranging from 35 to 85 years. Tumor size ranged from 1 to 17 cm at the largest diameter. Regional lymph node involvement was found in 4 (6.78%) cases. Moreover 1 case of pRCC showed subsequent metastasis during the follow-up period. Involvement of the adrenal gland was observed in one case of pRCC. Neither Gerota’s fascia nor renal pelvis involvement was observed in this study. The mean duration of follow-up was 47.88 months (SD = 27.81), with a median of 43.80 months (interquartile range = 25.25 to 66.22 months). The clinicopathological features of patients are summarized in Table 2 based on the subtypes of non-ccRCC.

### 3.2. Immunohistochemical Staining in Papillary RCC

A total of 41 pRCC cases were analyzed immunohistochemically for CD44, MMP-2, MMP-9, and Ki-67 expression in the neoplastic cells.

In the case of CD44 expression, 11 (26.83%) samples were negative, while upregulation of CD44 molecule was found in 30 (73.17%) cases.

Of all pRCC cases, 8 (19.51%) samples were negative for MMP-2 and 16 (39.02%) samples were negative for MMP-9.

Distributions of CD44, MMP-2, and MMP-9 expression based on overall immunohistochemical score are presented in Table 3.

The median value for Ki-67 was 4.5% (ranged from 1% to 8%).

### 3.3. Immunohistochemical Staining in Chromophobe RCC

A total of 18 chRCC cases were analyzed immunohistochemically for CD44, MMP-2, MMP-9, and Ki-67 expression in the neoplastic cells.

All chRCCs were positive for CD44. Additionally, all chRCCs, exept one, displayed MMP-2 expression, while 8 (44.44%) samples were negative for MMP-9.

Distributions of CD44, MMP-2, and MMP-9 expression based on overall immunohistochemical score are presented in Table 4.

The median value for Ki-67 was 4.0% (ranged from 0% to 9%).

### 3.4. Correlation of Clinicopathological Factors with the Expression of CD44, MMP-2, MMP-9, and Ki-67

In the case of pRCC, the expression levels of Ki-67, CD44, MMP-2, and MMP-9 were not associated with gender (*p* > 0.05, Mann–Whitney test), histologic nuclear grade, tumor size, and tumor necrosis (*p* > 0.05, Kendall’s rank coefficient). Only MMP-2 expression was positively correlated with patient age (τ = 0.31, *p* < 0.05, Kendall’s rank coefficient), while only CD44 expression was positively correlated with T (tumor) stage (τ = 0.26, *p* < 0.05, Kendall’s rank coefficient).

In the case of chRCC, the expression levels of Ki-67, CD44, MMP-2, and MMP-9 were not associated with gender and patient age, tumor size, and tumor necrosis (*p* > 0.05, Kendall’s rank coefficient). Ki-67 expression was negatively correlated with T stage (τ = −0.44, *p* < 0.05, Kendall’s rank coefficient), while the expression levels of CD44, MMP-2, and MMP-9 were not correlated with T stage (*p* > 0.05, Kendall’s rank coefficient).

### 3.5. The Expression Correlations between CD44, MMP-2, MMP-9, and Ki-67

In the pRCC group, CD44 expression positively correlated with MMP-9 (τ = 0.39, *p* < 0.05, Kendall’s rank coefficient), while did not correlate with MMP-2 and Ki-67 (*p* > 0.05, Kendall’s rank coefficient). MMP-2 and MMP-9 did not show a correlation with each other (*p* > 0.05, Kendall’s rank coefficient). Ki-67 expression did not correlate with either CD44 expression or MMP-2 and MMP-9 expression (*p* > 0.05, Kendall’s rank coefficient).

In the chRCC group, there were observed no correlations between Ki-67, CD44, MMP-2, and MMP-9 (*p* > 0.05, Kendall’s rank coefficient).

### 3.6. Prognostic Value of CD44, MMP-2, MMP-9, and Ki-67 Expression for Clinical Outcome

During follow-up, a death was documented in 6 (10.20%) patients—all these cases concerned the patients with pRCC (14.63%).

In the pRCC group, the results of statistical analysis demonstrated that the Ki-67 and CD44 expressions were not a significant factor affecting the overall survival (OS) of patients (*p* > 0.05). In the group with moderate and high MMP-2 expression, there was 100% OS. In the group with low MMP-2 expression, the OS was shorter (76.5%); however, the statistical analysis did not reach a significant result (*p* > 0.05). There was only a positive correlation between increased MMP-9 expression and shorter OS (*p* < 0.05). Patients with low MMP-9 expression show longer OS (93.5%) then patients with increased (moderate and high) MMP-9 expression (44.4% and 75.0% respectively). The shortest OS was observed in the group with moderate MMP-9 expression. All deaths were observed during the first 2 years after surgery (Figure 3).

There was no death in the chRCC group during follow-up, so it was not possible to carry out the statistical analysis in this group of patients.

## 4. Discussion

The subtypes of RCC are not only different histologically but are unique in their molecular or genetic profiles and tumorigenic mechanism. No reliable molecules have been described so far as clinically significant molecular markers in RCC. Numerous investigators reported the prognostic significance of CD44 overexpression in ccRCC, which is the most common type of renal cancer, but the role of this marker in non-ccRCCs is still not fully understood [9,17,26,27,28,29]. There are limited data regarding the expression of CD44 molecule in papillary and chromophobe RCCs, and as a rule, the number of cases is too small to achieve a relevant conclusion. CD44 protein is involved in several cellular functions, including cell–cell and cell–matrix adhesion, as well as migration, which are critical steps in cancer progression and metastasis. CD44 also binds to the extracellular matrix and acts as a platform to harbor growth factors and matrix metalloproteinases [28,29,30]. The present study was conducted to evaluate the expression and prognostic significance of CD44 in papillary and chromophobe RCCs. Our results contrasted with that of Heider et al. who found pRCC almost completely devoid of CD44 expression [31]. In our study, upregulation of CD44 molecule was found in 73.17% cases. Moreover, we proved that CD44 expression was positively correlated only with tumor stage in the case of pRCC, while there were no significant associations in the case of chRCC. In the study of Matusan et al. on 38 pRCCs, CD44s expression was positively associated only with lower pathological stage [26]. In contrast to these results, in the study conducted by Lee et al. in the non-ccRCC (20 pRCCs and 4 chRCCs) cases, CD44 expression was positively correlated with higher pathologic stage and histologic nuclear grade; however, multivariate analysis did not reach a statistically significant result for disease-free and overall survival [9]. In turn, Zanjani et al. observed no significant association was between CD44 expression and any important clinicopathological parameters and patients’ outcomes in 40 papillary and 30 chromophobe RCC cases [28]. Such a discrepancy in the results may be a consequence of a too-small number of analyzed cases to draw definite conclusions.

MMPs form a family of endopeptidases degrading the extracellular matrix and basal membrane. Furthermore, MMPs are involved in direct and indirect release of growth factors enhancing tumor growth and tumorigenicity. In particular, the ability to degrade type IV collagen, the major constituent of the basement membrane, is unique to MMP-2 and MMP-9. These two MMPs are most often linked to the malignant phenotype of tumor cells and are postulated to be a potential markers of tumor progression and prognosis [31,32,33,34]. The expression and involvement of several MMPs in RCC have been determined in several studies. However, the studies showed relatively conflicting results about their contribution to the clinicopathological findings and prognosis of the patients. Tissue MMP-2 and MMP-9 were found to be overexpressed in tumors and more frequently in non-ccRCC [35,36]. Lee et al. observed no correlations between MMP-2 and MMP-9 expression and gender, age, histologic nuclear grade, tumor (T) stage, nodal status, pTNM staging, and patient survival in non-ccRCC cases [9], which is in agreement with our results for chRCC, but not for pRCC. Kallakury et al. also analysed MMP-2 and MMP-9 expression in 20 pRCCs and 8 chRCCs by immunohistochemistry and demonstrated that increased expression of each protein correlates with high tumor grade; however, no correlation was found between increased expression of these proteins and advanced tumor stage. Moreover, only the increased expression of MMP-9 correlated with a shortened patient survival in their study, which corroborates our findings concerning pRCC [35]. Other researchers analysed MMP-2 and MMP-9 using techniques other than immunohistochemistry [37,38].

Additionally, Lee et al. assessed the expression correlations between CD44, MMP-2, and MMP-9 in the non-ccRCC group and revealed that CD44 expression did not correlate with either MMP-2 or MMP-9 expression, while MMP-2 and MMP-9 showed a positive correlation with each other [9]. In contrast to these results, our research showed a positive correlation between CD44 and MMP-9 expression, but we did not observe a correlation between MMP-2 and MMP-9 in the pRCC group. In turn, in the chRCC group, we observed no correlations between CD44, MMP-2, and MMP-9 expressions.

Ki-67 is a DNA-binding protein which is widely utilized as a marker of proliferation and detected by immunohistochemical staining [39]. There is a growing interest in Ki-67 as an effective molecular agent of the aggressive behaviour displayed by tumors and therapy response for survival outcome assessment in malignant neoplasms including RCC [39,40,41,42]. Some studies reported that Ki-67 immunohistochemistry is prognostically irrelevant in RCC patients and the function of Ki-67 in the prognosis of RCC remains inconsistent [43,44]. However, many studies have investigated role of proliferation index determined by Ki-67 as a powerful independent predictor in patients with RCC, but they either concerned only ccRCC or did not precisely specify the subtype of studied RCC. Only single studies determined the proliferation index of non-ccRCC subtypes using Ki-67. Mehdi et al. determined proliferation indices separately for seven pRCCs and six chRCCs. In their study, labelling index of Ki-67 in pRCC increased with tumor grade, but no such relationship was seen in the chromophobe renal tumours. Moreover, labelling index was significantly lower for chromophobe tumors Ki-67 = 2.1 ± 1.05% (median = 1.83%) as compared to pRCCs Ki-67 = 15.06 ± 14.25% (median = 10.79%) [8]. Ki-67 in our study was comparable in the case of pRCCs and chRCCs, which did not confirm the observation of Mehdi et al. Gontero et al. demonstrated that in the group of 46 pRCCs Ki67 index of >14% was a significant negative prognostic factor, but only in the univariate analysis [44]. These findings corroborate those described by Delahunt et al., who also found increased Ki-67 index to be independently associated with patient survival in the multivariate survival analysis for 68 cases of pRCCs [45]. Unfortunately, our study did not confirm the observation that Ki-67 can be a prognostic marker in the group of non-ccRCC.

## 5. Limitations

The primary limitation to the generalization of our results was the relatively small number of tumors included in this study due to the rarity of these neoplasms. For this reason, not all statistical tests normally requiring a larger sample size to ensure a representative distribution of the population could be used in our study. The subsequent limitation concerned the limited number of prior research on this topic. Most of the available studies concern ccRCC or do not specify the RCC subtype.

## 6. Conclusions

In summary, our study revealed that among the investigated markers only CD44 expression was positively correlated with advanced tumor stage, but exclusively in the group of pRCCs. Based on these findings, it could indicate a significant role of cancer stem cells in this group of neoplasms. Regarding clinical outcome, there was only a positive correlation between increased MMP-9 expression and shorter OS in the case of pRCCs, which emphasizes the role metalloproteinases activity in the mechanisms of carcinogenesis and may significantly improve the prediction of pRCC prognosis in the future. Taken together, it is not possible to unequivocally determine whether non-ccRCCs are more dependent on CSC or on MMPs activity.

To our best knowledge, it is the largest study investigating role of CD44, MMP-2, MMP-9, and Ki-67 as potential prognostic markers in the group of non-ccRCCs. Nevertheless, due to the rarity of these neoplasms, the number of tumors included in this study was still relatively small, and the final conclusion may not be transferable to the general population. Future analysis involving larger cohorts of patients could clear up if CD44, MMP-2, MMP-9, or Ki-67 are useful biomarkers for non-ccRCCs.

## Figures and Tables

**Figure 1 jcm-11-05196-f001:**
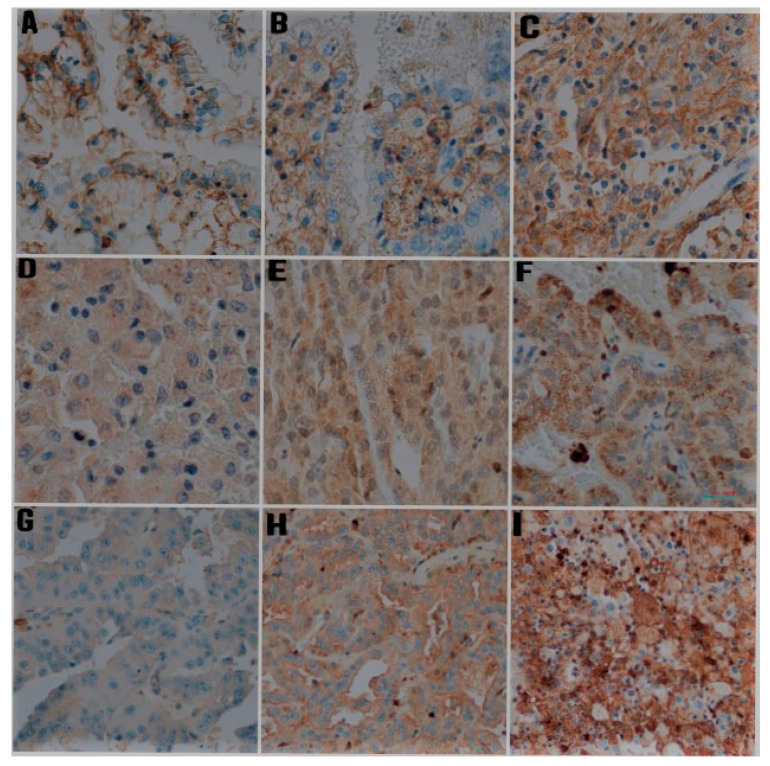
Representative photomicrographs of immunohistochemical staining for CD44, MMP-2, and MMP-9 in papillary renal cell carcinoma (IHC, ×400). (**A**) Weak staining intensity for CD44, (**B**) intermediate staining intensity for CD44, (**C**) strong intensity for CD44, (**D**) Weak staining intensity for MMP-2, (**E**) intermediate staining intensity for MMP-2, (**F**) strong intensity for MMP-2, (**G**) Weak staining intensity for MMP-9, (**H**) intermediate staining intensity for MMP-9, (**I**) strong intensity for MMP-9.

**Figure 2 jcm-11-05196-f002:**
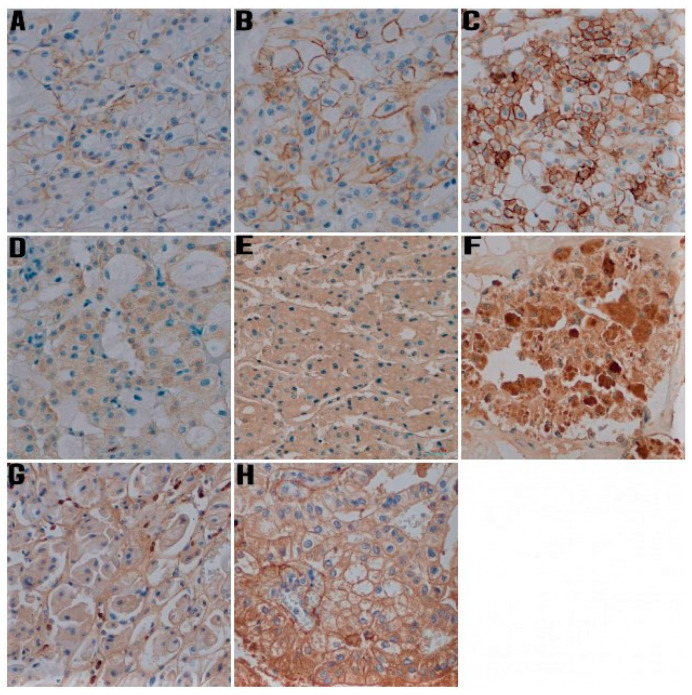
Representative photomicrographs of immunohistochemical staining for CD44, MMP-2, and MMP-9 in chromophobe renal cell carcinoma (IHC, ×400). (**A**) Weak staining intensity for CD44, (**B**) intermediate staining intensity for CD44, (**C**) strong intensity for CD44, (**D**) Weak staining intensity for MMP-2, (**E**) intermediate staining intensity for MMP-2, (**F**) strong intensity for MMP-2, (**G**) Weak staining intensity for MMP-9, (**H**) intermediate staining intensity for MMP-9.

**Figure 3 jcm-11-05196-f003:**
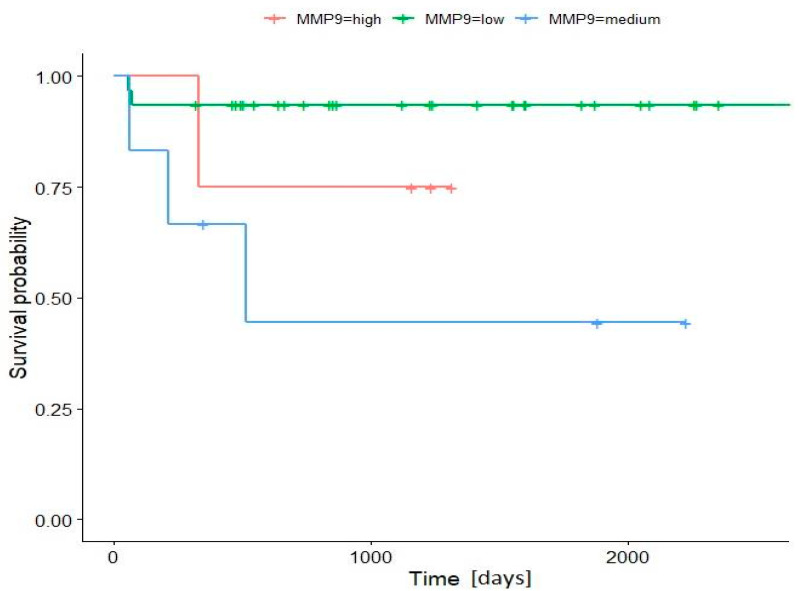
Kaplan–Meier survival curve according to MMP-9 expression. Patients with low MMP-9 expression show longer overall survival then patients with increased (moderate and high) MMP-9 expression (*p* < 0.05, Power = 0.96, Wilcoxon Gehan test with Mantel correction for multiple groups).

**Table 1 jcm-11-05196-t001:** Details of the primary antibodies used for immunohistochemistry.

Antibody	Clone	Source	Dilution	Incubation (min)
CD44	Monoclonal	MRQ-13, Cell Marque, Rocklin, CA, USA	1:300	20
MMP-2	Monoclonal	CA-4001, Zeta Corporation, Arcadia, CA, USA	1:50	60
MMP-9	Monoclonal	EP127, Bio SB, Goleta, CA, USA	1:100	60
Ki67	Monoclonal	MIB-1, Perlan, Beaverton, OR, USA	1:150	20

**Table 2 jcm-11-05196-t002:** Patients and tumor pathological characteristic of subtypes of non-clear cell renal cell carcinomas.

Histological Subtype of RCC	Papillary RCC	Chromophobe RCC	Total RCC	Power/ Effect Size	*p*
Number of tumor samples (*n* (%))	41 (69.50%)	18 (30.50%)	59 (100.00%)	-	-
Age, years (mean ± SD)	64.22 ± 9.62	60.83 ± 9.87	63.19 ± 9.74	d = 0.35 Power = 0.33	t = 1.24 *p* > 0.05
Gender (*n* (%))					
Female	9 (21.95%)	7 (38.89%)	16 (27.10%)	V = 0.18 Power = 0.27	Chi^2^ = 1.81 *p* > 0.05
Male	32 (78.05%)	11 (61.11%)	43 (72.90%)		
Type of operation (*n* (%))					
Radical nephrectomy	14 (34.15%)	6 (33.33%)	20 (33.90%)	V = 0.08 Power = 0.09	Chi^2^ = 0.04 *p* > 0.05
Partial nephrectomy	27 (67.85%)	12 (66.67%)	39 (66.10%)		
Tumor location (*n* (%))					
Right kidney	24 (41.46%)	9 (50.00%)	33 (55.90%)	V = 0.08 Power = 0.09	Chi^2^ = 0.37 *p* > 0.05
Left kidney	17 (58.54%)	9 (50.00%)	26 (44.10%)		
Tumor size, cm (mean ± SD)	4.9 ± 3.36	3.75 ± 2.24	4.55 ± 3.07	d = 0.40 Power = 0.40	t = 1.68 *p* > 0.05
Tumor stage (*n* (%))					
pT1	26 (63.42%)	11 (61.11%)	37 (62.70%)	V = 0.11 Power = 0.10	Chi^2^ = 0.69 *p* > 0.05
pT2	7 (17.07%)	2 (11.11%)	9 (15.30%)		
pT3	8 (19.51%)	5 (27.78%)	13 (22.00%)		
pT4	0	0	0		
WHO/ISUP grading				-	-
G1	11 (26.83%)	-	-		
G2	24 (58.54%)	-	-		
G3	2 (4.88%)	-	-		
G4	4 (9.76%)	-	-		
Tumor necrosis area % (mean ± SD)	14.51 ± 27.15%	0.56 ± 2.36	10.25 ± 23.50	d = 0.59 Power = 0.64	U = 404.50 *p* < 0.05
Sarcomatoid area % (mean ± SD)	2.46 ± 12.60	-	-	-	-
Rhabdoid area % (mean ± SD)	0	-	0	-	-
Lymphatic invasion present (*n* (%))	3 (7.32%)	0 (0.00%)	3 (5.10%)	V = 0.14 Power = 0.19	Chi^2^ = 0.28 *p* > 0.05
Angioinvasion present (*n* (%))	5 (12.20%)	0 (0.00%)	5 (8.50%)	V = 0.18 Power = 0.28	Chi^2^ = 1.08 *p* > 0.05
Neuroinvasion present (*n* (%))	2 (4.88%)	0 (0.00%)	2 (3.40%)	V = 0.12 Power = 0.15	Chi^2^ = 0.02 *p* > 0.05
Renal fibrous capsule invasion present (*n* (%))	19 (46.34%)	8 (44.44%)	27 (45.80%)	V = 0.04 Power = 0.06	Chi^2^ = 0.01 *p* > 0.05
Perinephric fat invasion present (*n* (%))	6 (41.63%)	4 (22.22%)	10 (16.90%)	V = 0.09 Power = 0.10	Chi^2^ = 0.11 *p* > 0.05
Renal sinus fat invasion present (*n* (%))	3 (7.32%)	6 (27.78%)	9 (15.30%)	V = 0.33 Power = 0.72	Chi^2^ = 4.69 *p* < 0.05
Renal sinus vascular invasion present (*n* (%))	3 (7.32%)	0	3 (5.10%)	V = 0.15 Power = 0.21	Chi^2^ = 0.29 *p* > 0.05
Dead (*n* (%))	6 (14.63%)	0	6 (10.20%)	V = 0.22 Power = 0.39	Chi^2^ = 1.55 *p* > 0.05

Legend: RCC—renal cell carcinoma, V—Cramer’s V, d—Cohen’s d, U—Mann–Whitney statistic.

**Table 3 jcm-11-05196-t003:** CD44, MMP-2, and MMP-9 immunoreactivity of papillary renal cell carcinoma.

Overall Immunohistochemical Score	CD44	MMP-2	MMP-9
Group 1 (low expression) (*n* (%))	28 (68.29%)	27 (65.85%)	31 (75.61%)
Group 2 (moderate expression) (*n* (%))	8 (19.51%)	11 (26.83%)	6 (14.63%)
Group 3 (high expression) (*n* (%))	5 (12.20%)	3 (7.32%)	4 (9.76%)

**Table 4 jcm-11-05196-t004:** CD44, MMP-2, and MMP-9 immunoreactivity of chromophobe renal cell carcinoma.

Overall Immunohistochemical Score	CD44	MMP-2	MMP-9
Group 1 (low expression) (*n* (%))	9 (50.00%)	7 (38.89%)	16 (88.89%)
Group 2 (moderate expression) (*n* (%))	5 (27.78%)	5 (27.78%)	2 (11.11%)
Group 3 (high expression) (*n* (%))	4 (22.22%)	6 (33.33%)	0 (0.00%)

## Data Availability

Not applicable.

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
