# Peer review of "Immunohistochemical Expression of CD44, MMP-2, MMP-9, and Ki-67 as the Prognostic Markers in Non-Clear Cell Renal Cell Carcinomas—A Prospective Cohort Study"

_jcm, 2022, doi:10.3390/jcm11175196_

Round 1

Reviewer 1 Report

The introduction is not a basic theory, but a practical demonstration of what has been researched so far.

The article does not take into account the existing articles on the topic under analysis. Examples:

https://wjso.biomedcentral.com/articles/10.1186/s12957-021-02268-5

https://onlinelibrary.wiley.com/doi/10.1111/j.1442-2042.2005.01159.x

These are just single examples!

The introduction is too sketchy and has little substantive value. The sentences are written too broadly. Example: “Many experimental and clinical studies prove that CD44 corresponds with a poor prognostic value [17−19].

There is a different way of writing p-value in this article. One time it is a specific value and another time p < 0.05.

The authors did not describe what is innovative in the research they conducted. A synthetic summary is missing.

There are too many uncertainties with the statistical analysis performed.

Table 1, 3 and 4 does not include the results of the relevant statistical tests.

There is no complete presentation of the obtained results in the form of figures.

In many places, the authors do not provide the exact results of the obtained statistical tests. Example: “Only 250 MMP-2 expression was positively correlated with patient age (p<0.05), ><0.05)…” The value of p alone (notated in different ways) is not enough.

There are no more advanced statistical methods used.

The size of the effect, measured with appropriate coefficients, should be calculated for individual tests (examples: Phi, Cramer’s V, Cohen’s, etc.).

Limitations related to the conducted study were not described.

Reviewer 2 Report

Aim of this study was to investigate the clinico-pathologic and prognostic values of the immunohistochemical expression of CD44, MMP2, MMP9, and Ki-67 in papillary and chromophobe RCCs. The study revealed that among the investigated markers only CD44 expression was positively correlated with advanced tumor stage, but exclusively in the group of pRCCs. Regarding clinical outcome, there was only a positive correlation between increased MMP-9 expression and shorter OS in the case of pRCCs. The paper has limitations which are correctly stated by authors. Would recommend to improve the quality of the figures, to better appreciate the positivity of the markers showed. The evolving scenario of the WHO classification introduced new histotypes and variants so the role of immunohistochemistry is much more important by the diagnostic pointy of view, please improve this part into your introduction or discussion.

Round 2

Reviewer 1 Report

The tables still do not contain the results of the appropriate statistical tests. Examples: 69.5% vs 30.5% or 4.9±3.6 vs 3.75±2.24, etc. (including effect size).

The most important results of the correlation analysis have not been presented in the form of figures.

The size of the effects has still not been calculated everywhere.

The limitations are very briefly described. There are many other limitations. The limitations should be a separate point.

Statistical test results are not recorded everywhere according to scientific standards, e.g. U = 23; p <0.05 (and effect size for each statistical test).
